# One-Dimensional Photonic Crystals Comprising Two Different Types of Metamaterials for the Simple Detection of Fat Concentrations in Milk Samples

**DOI:** 10.3390/nano14211734

**Published:** 2024-10-29

**Authors:** Mai Medhat, Cherstina Malek, Mehdi Tlija, Mostafa R. Abukhadra, Stefano Bellucci, Hussein A. Elsayed, Ahmed Mehaney

**Affiliations:** 1Physics Department, Faculty of Science, Beni-Suef University, Beni-Suef 62512, Egypt; mai.medhat@science.bsu.edu.eg (M.M.); ahmed011236@science.bsu.edu.eg (A.M.); 2Industrial Engineering Department, College of Engineering, King Saud University, P.O. Box 800, Riyadh 11421, Saudi Arabia; 3Materials Technologies and Their Applications Lab, Faculty of Science, Beni-Suef University, Beni-Suef 62512, Egypt; 4INFN—Laboratori Nazionali di Frascati, Via E. Fermi 54, 00044 Frascati, Italy

**Keywords:** hyperbolic metamaterials, gyroidal metamaterials, photonic crystal sensor, porosity, transfer matrix method, Tamm resonance, fat concentrations

## Abstract

In this study, we demonstrate the reflectance spectrum of one-dimensional photonic crystals comprising two different types of metamaterials. In this regard, the designed structure can act as a simple and efficient detector for fat concentrations in milk samples. Here, the hyperbolic and gyroidal metamaterials represent the two types of metamaterials that are stacked together to construct the candidate structure; meanwhile, the designed 1D PCs can be simply configured as [*G*(*ED*)*^m^*]*^S^*. Here, *G* refers to the gyroidal metamaterial layers in which Ag is designed in a gyroidal configuration form inside a hosting medium of TiO_2_. In contrast, (*ED*) defines a single unit cell of the hyperbolic metamaterials in which two layers of porous SiC (*E*) and Ag (*D*) are combined together. It is worth noting that our theoretical and simulation methodology is essentially based on the effective medium theory, characteristic matrix method, Drude model, Bruggeman’s approximation, and Sellmeier formula. Accordingly, the numerical findings demonstrate the emergence of three resonant peaks at a specified wavelength between 0.8 μm and 3.5 μm. In this context, the first peak located at 1.025 μm represents the optimal one regarding the detection of fat concentrations in milk samples due to its low reflectivity and narrow full bandwidth. Accordingly, the candidate detector could provide a relatively high sensitivity of 3864 nm/RIU based on the optimal values of the different parameters. Finally, we believe that the proposed sensor may be more efficient compared to other counterparts in monitoring different concentrations of liquid, similar to fats in milk.

## 1. Introduction

Milk is one of the most commonly consumed items in our daily diet, which is why it receives a focus from food safety organizations and researchers. Notably, milk is full of some essential nutrients, including vitamins, proteins, and minerals, which are crucial for maintaining good health at every stage of life [1]. Milk also contains remarkable quantities of ingredients which are important in the dairy products industry [2,3]. Natural milk that comes from cows, camels, and buffalos contains a lot of elements such as vitamin D, vitamin A, and calcium [4]. These elements have a significant effect on the health of muscles, teeth, and bones. Moreover, some minerals like zinc, sodium, phosphorus, potassium, and magnesium that support the health of the brain and heart and reduce the risks linked to obesity are also present in milk [2,4].

Meanwhile, the fat concentration in milk is essentially based on the concentrations of its constituents. Thus, the refractive index of milk changes with the variation in fat concentrations [5,6,7,8,9]. In this context, many researchers pay a lot of attention towards quantifying and discovering the adulterant materials in milk like added starch, sodium citrate, or formaldehyde, water, whey, formalin, sucrose, and vegetable oils [10]. Notably, these elements could have a significant effect on the fat concentrations in milk and its refractive index as well [10].

Therefore, the refractive index of milk could be crucial in determining the ratio of fat concentrations in milk. In this regard, Abb’s refractometer can be used to determine the refractive index of liquids; however, it is not relevant for opaque liquids like milk [11]. In addition, other types of refractometers may be not suitable for estimating the refractive indices due to many difficulties like adjusting the incident angle, the polarization mode, the size of the solid in liquid, and its time-consuming nature as well [12]. In recent times, there are some methods that have been developed to assess content in dairy product industries. These methods can be classified into three main categories: physical, chemical, and instrumental techniques [12,13,14,15,16].

The Röse-Gottlieb method is a one of these methods in which the detection of fats in milk depends on some organic solvent such as diethyl ether or petroleum ether, besides its complete dependence on a gravimetrical form [12,13]. Moreover, this method requires a considerable amount of both the sample and organic solvent, especially when the fat content in dairy products is low. Additionally, the sensitivity limitations of the weighing system can result in significant errors in the measured concentration of fats in milk samples. In contrast, Niklaus Gerber established the Gerber method, which is a straightforward method, providing rapid results, and is cost-effective as well. This technique has been designed for various dairy products, including whole milk, skimmed milk, and cream [13,14]. Despite the nonfat solids in the sample needing a concentrated sulfuric acid, the isolation of fat from the aqueous phase is achieved by adding amyl alcohol and then centrifuging the mixture. However, the Gerber method has several drawbacks. For example, it requires numerous specific instruments, including a butyrometer, pipette, centrifuge, and water bath, to conduct the test. Additionally, it cannot be automated and involves the risks associated with handling concentrated sulfuric acid. Through chromatographic techniques, there are two unique methods for detecting milk fat such as high-performance liquid chromatography (HPLC) and gas chromatography (GC) [15]. On one hand, the disadvantage of the GC method in estimating milk fat is focused on the instability of the milk sample during exposure to heat, which can be degraded as a gas. Although the HPLC method can successfully deal with thermally labile compounds, such as milk fat, the complicated software and hardware utilized in HPLC makes it expensive compared to other analytical tools. In this context, HPLC relies on specialized columns that are expensive and typically have a limited operational life.

On the other hand, Raman spectroscopy received considerable attention as a direct-rapid technique for measuring the amount of fat in milk [13,16]. In particular, the visible Raman spectroscopy depends on the relative change in Raman band intensity due to the total changes in the content of fat in milk [16]. However, the sensitivity of this method is limited due to the impact of some signals of other materials or components in milk, such as proteins and sugars, which, in turn, could contribute to the background during the test.

Due to these challenges, scientists and researchers recently adopted a new field of optical sensors based on photonic crystals (PCs) that could be promising through the detection and monitoring of many chemical, biological, and biomedical compounds.

Meanwhile, PCs have the ability to control the propagation of electromagnetic waves due to the nature of their construction, which, in turn, can facilitate a lot of significant effects in many applications [17,18]. In particular, the construction of PCs is mainly based on some homogeneous nanostructures of the periodic modulation of permittivity or refractive index in one, two, or three dimensions [19,20,21,22].

Moreover, the lattice constant of each unit cell for PCs should be comparable or compatible with the wavelengths of incident light for guiding light through narrow and specific channels [16]. Therefore, PCs provide some stop bands at certain frequencies of electromagnetic waves named photonic band gaps (PBGs) [23,24]. These PBGs are capable of prohibiting or preventing the propagation of certain frequencies of the incident electromagnetic waves due to Bragg’s scattering at the interferences [19].

Nowadays, with the escalating improvement of PCs’ properties, PCs have become the mainstay of many applications such as sensors, diode lasers, and optical switches as well [25,26,27]. Notably, PCs provide variety in designs and fabrications based on some different types of materials such as metals, semiconductors, superconductors, 2D materials, nanocomposites, and metamaterials (MMs) [28,29].

Over the past two decades, MMs received tremendous attention in the design and construction of PC structures due to their superior optical response. MMs are types of artificial materials whose magnetic permeability and electric permittivity tensors do not have the same sign [30,31]. These materials are recalled as left-hand materials due to their negative refractive index. Also, MMs are widely utilized in many optical and physical applications such as lenses, absorbers, cloaking, and sensors as well [27,32]. Basically, the characteristics such as orientation, geometry, and precise of MMs were discovered by Veslago in 1968 [28,29]. However, 32 years later, Smith et al. demonstrated the experimental validation of these materials in 2000 [30]. Recently, researchers have directed their attention towards some new designs of indefinite MMs like gyroidal MMs (GMMs) and hyperbolic MMs (HMMs) [33,34,35].

In this regard, HMMs have received a lot of attention as a kind of strongly anisotropic MMs in many applications [32,33,34,35]. HMMs were given their name because of the hyperboloidal nature of their iso-frequency dispersion surface [34]. HMMs like nanocomposite structures can be constructed based on dielectric–dielectric or metallic–dielectric multilayers that provide a dispersive loss nature lower than bulk metallic structures [36,37]. Therefore, HMMs provide different signs in the orthogonal directions of their permittivity. Moreover, HMMs have high k-modes that dominate the diffraction limit conversely in conventional materials [38,39,40]. Meanwhile, PC structures comprising HMMs are capable of introducing a cavity mode inside the dispersionless gap or band whatever the angle of incident [37]. Therefore, HMMs have distinct and unique aspects in different applications such as super lenses, selectors, and perfect absorbers [41,42,43].

In contrast, GMM is also a new kind of MM that could be constructed due to the inclusion of a thin layer of metal in a gyroidal configuration form inside a hosting dielectric medium in a 3D configuration [44]. In addition, GMM has a distinguishable average curvature close to zero due to its design in tri-helix geometry and its real part has a negative permittivity [45,46]. Indeed, the experimental measurements emphasize the process of fabrication in the periodicity of dielectric layers filled with metallic portions [47]. Notably, the GMM configuration has attracted attention from researchers due to its hollow form structure as in butterflies’ wings [42,48]. These features make GMM a promising design in many applications such as for switches, thermal emitters, organic solar cells, and sensors as well [44,45,49,50,51,52].

Therefore, we believe that the design of the PC structure based on both types of MMs, HMMs and GMMs could be of significant interest in the optics and photonics communities. Meanwhile, we introduced in the present study a 1D PC comprising HMMs and GMMs to act as a sensor for the detection of fat concentrations in milk. The designed 1D PCs can be simply configured as [*G*(*ED*)*^m^*]*^S^*. Here, the symbol *G* refers to the GMM layers in which Ag is embedded in a gyroidal configuration of TiO_2_. In contrast, (*ED*) defines the HMM layers in which two layers of porous SiC (*E*) and Ag (*D*) are combined together with *m* number of unit cells. The porous silicon carbide (SiC) layers, with a specific porosity ratio of 0.20, were filled with milk fat samples of different concentrations. The detection procedure primarily relies on the emergence of three resonant peaks at specified wavelengths ranging from 0.8 μm to 3.5 μm. However, the first peak located at 1.025 μm is favored in the detection process due to its low reflectivity and narrow full width. The position of this peak is strongly affected by the changes in the concentrations of milk samples through the voids of SiC layers. Meanwhile, a maximum sensitivity of 3864 nm/RIU is attained at a 1.5% concentration of fats in the sample. Therefore, we believe that our proposed sensor could be promising compared to some of its similar counterparts [2,5,53]. For example, in 2021, Khedr M. Abohassan et al. proposed a binary PC design to detect the fat concentration in milk [5]. Meanwhile, the maximum sensitivity of this design is about 150.9 nm per RIU. Then, in 2022, Abdulkarem H. M. Almawgani et al. demonstrated the detection of fat concentrations in milk using 2D material like Mxene (Ti_3_C_2_T_x_) with a thin layer of silicon and silver based on the emergence of surface plasmon resonance [2]. Meanwhile, the sensitivity of this work reaches 350 deg/RIU. In the same year, Zaky A. Zaky et al. introduced a 1D ternary PC structure of porous Si layers with a sensitivity of 2925 nm/RIU to detect the concentrations of fats in milk through the defect layer of the designed structure [53]. Most of the previous designs regarding the detection of some fluidic components based on the 1D PC sensors mainly depend on the inclusion of a defect layer through the designed PC structure to generate the resonant mode [2,3,53]. For other types of PC sensors, a defect layer can be introduced between a thin metallic layer and the designed PC structure or multilayer structure to generate Tamm plasmon (TP) resonance or surface plasmon resonance [17]. Here, the presence of GMM could give rise to the emergence of TP resonance without including any thin metallic layer on the top surface of the designed PCs. Meanwhile, our design is free of any defect layer because the analyte material can smoothly flow through the voids present within the HMM layers. In particular, the emergence of the defect layer could provide some mismatching during the manufacturing procedure. In addition, it can also reduce the level of interaction between the incident radiation and the designed structure besides the limited values of the overall performance. Therefore, we focus on two different metamaterials: GMM and porous HMM, eliminating the need for a defect layer. This approach helps in avoiding the mismatches and manufacturing difficulties, while it can also improve the accuracy of the detector due to the promising optical properties of MMs. Additionally, the optical properties of GMM and HMM enable the demonstration of optical Tamm plasmon (TP) resonance without the need for a prism or two-dimensional materials. Furthermore, the superiority of our detector is evident in its performance, achieving a maximum sensitivity of 3864 nm per RIU, without the need for excessive layers or complex manufacturing processes that may drive up costs. Moreover, our numerical outcomes and findings demonstrate that our design can be highly acceptable in the milk sensing process. As far as we know, no previous study used HMM as a sensor for milk or similar liquids so this innovative type of sensor can be used in dairy industries.

In what follows, we introduce theoretical methodology and model design. Here, the effective medium theory, characteristic matrix method, Drude model, Bruggeman’s approximation, and Sellmeier formula represent the mainstay of our theoretical analysis and methodology.

## 2. Theoretical Methodology and Design

Figure 1 presents a schematic diagram of designed 1D PCs surrounded by air and the substrate. Here, the GMM layers are specified with refractive index $n_{G}$ and thickness d_1_. In contrast, the HMM layers are characterized by refractive index $n_{H}=\sqrt{\varepsilon_{H}}$ and thickness d_H_ = d_2_ + d_3_. As previously mentioned, each layer of the HMMs is designed as a composite structure of porous SiC of thickness d_2_ and refractive index n_P_ besides the Ag metal of thickness $d_{3}$ and refractive index n_D_. Also, the 3D schematic diagram in Figure 1 indicates that the reflected electromagnetic waves from the designed structure can be amplified and analyzed across the signal amplifier and signal analyzer, respectively. As shown in Figure 1, each unit cell of the proposed PC sensor can be designed by the deposition of a GMM layer over *m* layers of HMM. Then, the whole structure can be investigated in the form of *S* number of unit cells to be configured as [*GMM* (*HMM*)*^m^*]*^S^* or ${\left[(Ag/TiO_{2}){\left(P(SiC)/Ag\right)}^{m}\right]}^{S}$. Meanwhile, the milk samples of different fat concentrations can be deflected towards the porous SiC voids to be detected. In particular, the detection procedure is mainly based on the interaction of the incident electromagnetic waves with the designed PC structure. Notably, this interaction could give rise to the emergence of some resonant peaks through the PBG formed. In addition, the characteristics of these resonant peaks, such as their position, intensity and full width at half maximum, as well are very sensitive to the changes in the concentrations of the milk samples. Therefore, the detection process can be smoothly investigated based on the properties of such resonant peaks.

Now that we have introduced the topic, in what follows the strategy at which the incident electromagnetic waves can interact with our designed 1D PC structure will be shown. In the vicinity of the effective medium theory, the Drude model and the optical properties of the considered materials, the transfer matrix method (TMM) can offer a simple solution to describe such an interaction [39,51,52,53,54,55]. Meanwhile, the designed structure is considered to be periodic through the *Z*-axis. Then, for the case of the normal incidence of interacting electromagnetic waves, TMM can describe the response of these waves through each layer of the designed structure in a matrix form. Thus, the electric and magnetic field components of these waves via the smallest unit in our structure can be written as follows:(1)$$E_{i}=F_{i}\exp\left(-ik_{i}x\right)+G_{i}\exp\left(ik_{i}\;x\right)=E_{x^{+}}+E_{x^{-}}\;H_{i}=\frac{\partial E_{i}}{i\omega \;\partial x}{=\frac{k_{i}}{\omega }[-F}_{i}\exp\left(-ik_{i}x\right)-G_{i}exp(ik_{i}x)]=q_{i}\;\left(E_{x^{+}}-E_{x^{-}}\right)$$
where $\;F_{i}$ and $G_{i}$ are the field amplitudes in layer  i, λ is the wavelength of incident waves, $k_{i}$ is the wave vector in layer  i, and it is formed as $\;k_{i}=k_{0}n_{i}cos\theta_{i}=(2\pi /\lambda )\;n_{i}cos\theta_{i}$, $\;\theta_{j}$ and $n_{j}$ depicts the angle of incidence and the refractive index through this layer, respectively. In other formulism, the previous equation could be written as [54,55] follows:(2)Ei Hi =11qi−qiEx+Ex−

For a distinct layer i with thickness $d_{i}=z_{1}-z_{0}$, the resultant response of the electromagnetic waves between the boundaries (*z*_0_ and *z*_1_) appears in the following form:(3)E0H0 =11qi−qiexp⁡i kidi00exp⁡i kidiEx1+Ex1−           =cos⁡(kidi)(−i/qi)sin⁡(kidi)−iqisin⁡(kidi)cos⁡(kidi)E1H1 =wiE1H1 

Then, the matrix *w*_i_ is utilized to describe the response of the incident waves through each layer of the designed structure along Z-axis, such that [55]
(4)$$w_{i}=\left(\begin{array}{cc} \cos\left(q_{i}\right) & (-i/\varsigma_{i})\sin\left(q_{i}\right)\\ {-i\varsigma }_{i}\sin\left(q_{i}\right) & \cos\left(q_{i}\right) \end{array}\right)$$
where the components $q_{i}$ and $\varsigma_{i}$ for TM polarization equal $\frac{2\pi d_{i}}{\lambda }\cos\theta_{i}/n_{i}$ and $\frac{\cos\theta_{i}}{n_{i}}$, respectively. For TE, we have $q_{i}$ = $\frac{2\pi d_{i}}{\lambda }n_{i}\cos\theta_{i}$ and $\varsigma_{i}=n_{i}\cos\theta_{i}$. Then, the total characteristic matrix that is obtained from the response of our candidate sensor to the electromagnetic waves can be written in the following form [41,56]:(5)$$w_{t}=W=\left(\begin{array}{cc} W_{11} & W_{12}\\ W_{21} & W_{22} \end{array}\right)=\prod_{i=1}^{k}w_{i\;}\;,i=G,E,D\;={\left(W_{G}W_{H}\right)}^{S}={\left({W_{G}(W_{E}W_{D})}^{m}\right)}^{S}$$

Here, $w_{t},W_{G},W_{H}$ are the matrices of the whole structure, GMM layer, and HMM layers, respectively. Consequently, Equation (5) can be introduced to describe the reflectance coefficient of the constructed design as [57,58] follows:(6)$$r_{1}=\frac{\left(W_{11}+W_{12}\varsigma_{s}\right)2\varsigma_{0}-{(W}_{21}+W_{22}\varsigma_{s})}{\left(W_{11}+W_{12}\varsigma_{s}\right)\varsigma_{0}+(W_{21}+W_{22}\varsigma_{s})}$$

Finally, the reflectivity of our sensor are computed as [59,60] follows:(7)$$R=\left|\left.{r_{1}}^{2}\right|\right.$$

Then, the effective medium theory is utilized to investigate the optical characteristics or the permittivity of the HMMs [21,56]. Meanwhile, this theory investigates theoretically the permittivity of HMM in a matrix formulism, such that [2,25,61]
(8)$$\varepsilon_{H}=\left(\begin{array}{ccc} \varepsilon_{Hx} & 0 & 0\\ 0 & \varepsilon_{Hy} & 0\\ 0 & 0 & \varepsilon_{HZ} \end{array}\right)$$

Here, the permittivity of HMM elucidates in $\varepsilon_{H}$, while $\varepsilon_{HZ},\varepsilon_{Hx}$ defines the vertical and parallel components of the permittivity of HMM, respectively. Here, these components are outlined in the following form [30,38,62]:(9)$$\varepsilon_{HX}={h\varepsilon }_{E}+\left(1-h\right)\varepsilon_{D}$$
(10)εHZ=εEεDεDff+εE(1−h)

Hence, $\varepsilon_{E}$, $\varepsilon_{D}$ describe the permittivity of the porous SiC and Ag layers, respectively. Hence, the vertical and parallel components of HMM are mainly dependent on the filling ratio (h). Here, this factor is expressed in terms of the thicknesses of porous SiC and Ag layers $\left(d_{E},d_{D}\right)$ such that [63]
(11)$$h=\frac{d_{E}}{d_{E}+d_{D}}$$

Then, for the effective refractive index of porous SiC, Bruggeman’s effective medium approximation (BEMA) is a suitable method to study the effect of porosity on the effective refractive index [64,65]:(12)$$p\left(\frac{\left({n_{V}}^{2}-{n_{eff}}^{2}\right)}{\left({n_{V}}^{2}+2{n_{eff}}^{2}\right)}\right)+\left(1-p\right)(\frac{\left({n_{SiC}}^{2}-{n_{eff}}^{2}\right)}{\left({n_{SiC}}^{2}+2{n_{eff}}^{2}\right)})=0$$

Here $n_{Sic}$, $n_{V}$ and $n_{eff}$ are the refractive indices of SiC, the material embedded inside the pores, and the effective refractive index of the porous SiC layer, respectively. Then, we can express the effective refractive index of porous material as follows:(13)$$n_{eff}=\frac{1}{2}{\left(Z+\sqrt{Z^{2}+8{n_{SiC}}^{2}{n_{V}}^{2}}\right)}^{\frac{1}{2}}$$
(14)$$Z=3p\left({n_{V}}^{2}-{n_{SiC}}^{2}\right)+(2{n_{SiC}}^{2}-{n_{V}}^{2})$$
where the empirical relationship between the refractive index of SiC and the incident wavelengths can be demonstrated based on the Sellmeier approximation formula as [66,67] follows:(15)$$n_{SiC}=\sqrt{\frac{0.20075\lambda^{2}}{\lambda^{2}+12.07224}+\frac{5.54861\lambda^{2}}{\lambda^{2}-0.02641}+\frac{35.65066\lambda^{2}}{\lambda^{2}-1268.24708}+1}$$

Next, the Drude model is considered a suitable way to identify the permittivity of metal (Ag), such that [56]
(16)$$\varepsilon_{D}=1-\frac{{\left({ϣ}_{p}\right)}^{2}}{\omega^{2}+i\omega \Upsilon }$$

Here, ${ϣ}_{p}$ and Υ are denoted as the Plasmon and damping frequencies, respectively. Then, these constants for Ag have the values ${ћ\;ϣ}_{p}=9.01\;eV$, and ћ Υ=0.048 eV with ћ as Planck’s constant.

For the GMM layers, the permittivity is highly dependent on several factors, including the gyroidal geometry, the refractive index of the host medium, and the refractive index of the metal embedded within the gyroidal unit cells [63,64,65,66,67,68,69,70]. In this regard, TiO_2_ is chosen as a hosting dielectric medium of the gyroidal layer. Therefore, the refractive index of TiO_2_ $\left(n_{h}\right)$ as a hosting material through the Gyroidal layer is given as [68,69] follows:(17)$$n_{h}={\left(5.913+\frac{0.2441}{\lambda^{2}-0.0803}\right)}^{\frac{1}{2}}$$

Furthermore, the permittivity of the GMM ($\varepsilon_{G}$) can be expressed as [70,71,72,73] follows:(18)$$\varepsilon_{G}=1.193\left(A-B{\left(C\right)}^{2}\right)$$
where *A*, *B*, and *C* are different parameters that can strongly tune the permittivity of the GMMs such that
(19)$$\left\{\begin{array}{c} A=\sqrt{2}\\ B=16\sqrt{2\;}Q^{2}\\ C=\frac{\pi \sqrt{-\varepsilon_{D}}}{2\sqrt{2}\;n_{h}}-1 \end{array}\right.$$
(20)$$r_{A}=0.29\;r_{a}\sqrt{ff},\;\lambda_{A}=1.15\;r_{a}D,\;D=\sqrt{1-(0.65ln(ff)})$$

In Equation (18), Q=(rAλA)2 is a fundamental term related to the geometrical constant with $r_{A}$ to denote the radius of the Gyroidal helix, which is known as a function of the filling fraction of the used metal Ag (ff) and helix length $r_{a}$. $\lambda_{A}$ represents an effective geometrical parameter that is known as the effective plasmon frequency of an ideal gyroid and can be expressed as a function of the filling fraction of the metal embedded inside the GMM layers. In addition, $\varepsilon_{D}$ is known as the permittivity of metal, as shown in Equation (16).

Finally, we clarify the dependence of the fat concentration on the refractive index of the milk sample $\left(n_{V}\right)$. Meanwhile, we plot in Figure 2 the change in the refractive index of the commercial milk and the increase in the fat concentration [5,74]. The figure shows that the refractive index increases gradually with the fat concentration. This response can be described based on the quadratic fitting of these experimental data. Meanwhile, the following relation can be introduced to express the refractive index of the commercial milk due to the increase in its fat concentration from 0 to 33.3%:(21)$$C_{f}\left(\%\right)=14\times 10^{3}{n_{V}}^{2}-{37\times 10^{3}n}_{V}+25\times 10^{3}\;(quadratic\;fitting)$$

## 3. Results and Discussion

### 3.1. Optimization of Parameters

In this section, we introduce the numerical findings of the considered sensor based on the theoretical treatment of the designed 1D PC structure. The design of the candidate sensor is configured as ${\left[(Ag/TiO_{2}){\left(\;P(SiC)/Ag\right)}^{4}\right]}^{2},$ as shown in Figure 1. In all calculations, the thicknesses of the GMM and HMM layers are considered to be $d_{1}$ and ${\;d}_{H}$, respectively. In the beginning, we contemplate that $d_{1}=40\;nm$, $d_{H}=d_{2}+d_{3}$, and the incident angle $\theta_{0}=0{}^{\circ}$. Here, the periodicity of our 1D PCs (S) is equal to two, which, in turn, could be promising through the fabrication process compared with its counterparts in PC design. Then, the voids in porous SiC were filled with the milk samples to detect the different concentrations of fats in these samples. Then, we turned our attention towards the optimization procedure of the different parameters related to the designed structure to investigate the highest performance of the designed sensor regarding the optimum values of these parameters. In fact, the optimization of the constituent materials represents a crucial step towards providing the optimum performance of the considered sensor. Notably, the detection process is essentially related to the response of the electromagnetic waves through the designed structure due to the emergence of some resonant peaks through the reflectance spectrum of this designed structure. Therefore, the optimization of some related parameters such as thicknesses, the porosity of SiC layers, the filling fraction of gyroidal Ag, filling ratio of HMM layers and periodicity as well could have some significant effects on the properties of the emerged resonant peaks and the performance of the designed sensor as well. In particular, the changes in the values of these parameters could give rise to some distinct variations in the optical path length of the interacting electromagnetic radiation, which, in turn, leads to some changes in the position, and intensity of the emerged resonant peak. In this regard, the upcoming subsections simulate in detail the optimization process of these parameters towards the optimum performance of the designed sensor.

#### 3.1.1. Optimization of Porosity

Firstly, we introduce in Figure 3 the role of porosity for porous SiC layers on the performance of the designed sensing tool at different concentrations of the milk samples. We believe that the value of porosity could give rise to a significant effect on the overall performance of the designed sensor. In particular, the change in the porosity of SiC layers provide a distinct effect on the effective permittivity of the SiC layer, as investigated in Equation (13). In other words, the increase in the porosity values or the number of voids could decrease the effective refractive index of the porous SiC layer [53]. In addition, this effect is not only investigated on the effective permittivity of porous SiC but also includes both vertical and parallel components of HMM’s permittivity, as listed in Equations (9) and (10). Thus, the permittivity of HMM could provide some significant changes, which, in turn, may be effective on the optical path length of the incident radiation and the characteristics of the resonant peak as well. Therefore, some effects on the performance of the designed sensor can be introduced. In this regard, the change in the porosity’s values of porous SiC leads to significant changes in the performance of the designed sensor based on its sensitivity values, as shown in Figure 3. Here, the change in the porosity values from 0.2 to 0.6 produces some decrements in the sensitivity of the proposed sensor over all the fat concentrations of milk samples. Specifically, changes in the porosity values (*p*) lead to variations in the effective refractive index of the porous silicon carbide (SiC) layers. The effective refractive index decreases from 2.166 to 1.832 as porosity increases from 0.2 to 0.6. This reduction has a significant impact on the behavior of propagating electromagnetic waves and the overall performance of the designed sensor. Consequently, the optimal porosity value that results in the highest sensitivity is 0.2. Additionally, this value is noteworthy in the manufacturing process due to its robustness against mechanical stress.

#### 3.1.2. Optimization of Filling Fraction

In this subsection, we demonstrate the optimum value of the Ag filling fraction through the GMM layers. The optimization procedure of this parameter is mainly introduced at the optimum value of SiC porosity i.e., *p* = 0.2. As depicted in Equations (18)–(20), the filling fraction can lead to some changes in the permittivity of the GMM, which, in turn, can change the response of the incident radiation and the performance of the designed sensor as well. Meanwhile, Figure 4 discusses the role of the Ag filling fraction on the sensitivity of the designed sensor. In this regard, a little value of *ff* = 0.08 leads to a maximum sensitivity of 3864 nm/RIU, as shown in Figure 4. For other values smaller or greater than 0.08, the sensitivity of the considered sensor decreased significantly, especially at the low-fat concentrations of milk samples. Therefore, the inclusion of Ag with *ff* = 0.08 represents the optimum value towards the maximum value of sensitivity. Moreover, this value could be of potential interest towards better performance in the vicinity of the low full width at half maximum and high-quality factor as well. In particular, this little value of the Ag filling fraction could give rise to little absorption values of the incident electromagnetic waves.

In contrast, we considered the optimum values of both SiC porosity and the Ag filling fraction to discuss the optimum value of the filling ratio through the HMM. Meanwhile, the increase in the filling ratio of the HMM layers led to significant decrements in the sensitivity of the design considered, as shown in Figure 5. Here, the variation in the value of h from 0.40 to 0.50, 0.60, and 0.70 decreased the value of sensitivity from 3864 nm/RIU to 2273 nm/RIU, 1364 nm/RIU, and 1136 nm/RIU, respectively. In particular, the change in the filling ratio led to some changes in the thicknesses of the SiC and Ag that presented the building block of the HMM layers, as elucidated in Equation (11). Moreover, both vertical and parallel components of HMM’s permittivity were directed towards some changes. Therefore, the optical path length of the incident electromagnetic radiation through the designed structure gave rise to some significant changes due to the variations in both HMM’s permittivity and the thicknesses of its constituent materials as well. In this regard, some crucial variations in the optical characteristics of the resonant peak and the sensor performance were expected to appear, as shown in Figure 5. Accordingly, a filling ratio of 0.4 represents the optimum value that provides the maximum value of sensitivity.

#### 3.1.3. Optimization of Thickness

Based on the optimum values of the previous parameters, we turn our attention here to investigate the optimum value for thickness. In particular, the changes in the thickness of the different layers provide significant effects on the overall optical properties of the designed structure. However, we limited our discussion only to the impact of the thickness on the thickness of the GMM’s layers. Notably, the role of the variation in the thickness of the HMM’s layers is included through the discussion of the filling ratio role. Therefore, we have now discussed the optimal thickness of the GMM’s layers, as shown in Figure 6. A thickness of 40 nm was identified as the optimal value for the GMM layers due to the maximum sensitivity observed at this thickness. Figure 5 demonstrates that increasing the thickness of the GMM layers results in a reduction in sensitivity values, attributed to weakly localized states. Specifically, when the thickness of the GMM layers ranges from 40 nm to 60 nm, sensitivity decreases from 3864 nm/RIU to 681.8 nm/RIU, passing through a value of 1136 nm/RIU at the lowest refractive indices of the milk samples.

#### 3.1.4. Optimization of Periodicity

Based on the optimum value of the different optimized parameters in which *p* = 0.20, ff=0.08, h=0.4, $d_{1}=40\;nm$, $d_{2}=80\;nm$, and $d_{3}=120\;nm$, we discuss here the sensitivity of our proposed sensor regarding the periodicity of the whole structure and that of the HMM as well. In this regard, the designed structure is constructed as ${\left[(Ag/TiO_{2}){\left(P(SiC)/Ag\right)}^{m}\right]}^{S}$ including a specific number of periods *m* for the HMM layers and *S* for the whole structure. Figure 7 investigates how the maximum sensitivity decreases with *m* and *S* both increasing, respectively. For a fixed number of *S* = 2, Figure 7a demonstrates that the best response for the sensitivity values over all the considered fat concentrations of milk samples can be obtained at *m* = 4. In particular, HMM as a new kind of MM provides some peculiar optical properties due the changes in its periodicity. Notably, the change in its periodicity gives rise to the change in its overall thickness, which, in turn, has some distinct effects on the optical properties of the designed structure [38,39,40,41]. In this regard, the increase in *m* values may lead to the shrinking of the full width at half maximum of the resonant peak. This behavior could improve the performance of the sensor regarding its figure of merit and quality factor as well. However, the sensitivity may suffer from some decrements because of the weak shift in the resonant mode. In contrast, at the optimal periodicity of the HMM, the highest sensitivity is achieved when the periodicity number of the entire structure equals two, as illustrated in Figure 7b. Specifically, the design of a PC sensor with a large number of periods makes the edges of the PBG besides the resonant peaks become sharper and steeper [53]. This response may improve the performance of the sensor regarding its figure of merit and quality factor as well because the increase in the periodicity number of PCs gives rise to the shrinking of the full width at half maximum of the resonant peak [2,5,53]. Thus, a smaller value of *m* and *S* enhances the interaction between light and the proposed material structure, leading to the improved efficiency and optimal performance of our detector. Furthermore, these values may become increasingly significant during the manufacturing process. Notably, the manufacturing of the PC structure with a limited number of unit cells to act as a sensing tool could be of significant interest due to the limited cost, relatively high accuracy and sensitivity as well compared to those of large numbers of periods.

#### 3.1.5. Optimization of Incident Angle

Finally, we introduce in this subsection the role of the angle of incidence on the performance of our sensor at the optimum values of the different parameters. Notably, the incident angle can also provide some effects on the sensitivity and the position of the resonance peak for the candidate design. Figure 8 illustrates the impact of the incident angle on the sensitivity of our sensor ${\left[(Ag/TiO_{2}){\left(P(SiC)/Ag\right)}^{4}\right]}^{2}$ at different fat concentrations. The variation in the incident angle $\theta_{0}$ from 0 to 80 degrees for TE polarization shows a distinct variation in the values of sensitivity, especially at low concentrations. Meanwhile, increasing the incident angle to 80 degrees at $n_{V}=1.3496$ results in decreasing sensitivity from 3864 nm/RIU to 1523 nm/RIU, 1295 nm/RIU, 1136 nm/RIU, and 1091 nm/RIU. Due to its compatibility with other parameters like the thickness and filling ratio, the case of normal incidence provides the maximum value of sensitivity over all the considered concentrations. In particular, this parameter has a significant effect on the optical path length in parallel with the thicknesses of the considered layers and the wavelength of the incident radiation as well.

### 3.2. Sensor Analysis

Now, we introduce in this subsection the overall performance of the considered 1D PC sensor based on the optimum values of the geometrical and structural parameters of the constituent materials. To sum up, the optimal values of the different parameters can be elucidated as *p* = 0.20, ff=0.08, h=0.4, $d_{1}=40\;nm$, $d_{2}=80\;nm$, $d_{3}=120\;nm$, *m* = 4, *S* = 2 and $\theta_{0}=0{}^{\circ}$. Now, we can comprehend the optical properties of the reflectance spectrum along the direction of incident wavelengths, as shown in Figure 9. This figure shows the emergence of three resonant peaks at different spectral positions, i.e., $\lambda_{res1}=1.025\;\mu m,\;\lambda_{res2}=1.532\;\mu m$, and $\lambda_{res3}=2.96\;\mu m$, respectively. In fact, these resonant peaks belong to TP resonance. Such a type of resonance can be excited due to the deposition of a thin metallic layer over a periodic multilayer structure like PCs [75,76,77]. TP resonance is well-known as a normal-incidence- excitation due to the generation of asymmetric metal–dielectric modes [68]. Moreover, the range of the transmission peak (TP) resonance modes is primarily determined by the optical dispersion characteristics. TP resonance has been extensively discussed in the literature due to its exceptional optical properties, which could be advantageous in various applications, including sensing [78,79,80,81]. Notably, TP resonance is distinguished by its high sensitivity and quality factor in response to various chemical, physical, and biological changes. Additionally, the presence of a wide PBG enhances the stability of the TP state without coupling to other resonant modes [17]. Recently, TP resonance has been used in photo detection and in thermal emission field, as illustrated in the study by Want et al. [77,78,79].

However, most of the previous designs related to the emergence of TP resonance for sensing applications basically depend on the presence of a defect layer to be filled with the considered analyte. Here, our designed structure is free of any defect layer, which could be of significant interest through the manufacturing procedure. Meanwhile, we have considered through our detection procedure the resonant mode $\lambda_{res1}=1.025\;\mu m$ due to its high sensitivity against the variation in fat concentrations besides its low values of both reflectivity and full width at half maximum as well. Therefore, we believe that this resonance could provide the best performance in the vicinity of high sensitivity, good quality factors, and low detections limit as well compared with the others.

Consequently, Figure 10 illustrates the response of the resonant peak to variations in the fat concentrations (*C_f_*) of milk samples. The figure shows a redshift in the transmission peak resonance as the concentration increases. Notably, the resonant peak shifts significantly despite only minor changes in fat concentrations. This shift occurs due to alterations in the optical path length of the incident radiation, as influenced by the fat concentration. Specifically, changes in fat concentrations have a considerable impact on the effective refractive index of porous silicon carbide (SiC) layers, as demonstrated in Equations (12) and (13). Meanwhile, the numerical findings in this figure can give an indication for the performance of the considered sensor.

In the following, our sensor’s efficiency and performance are evaluated not only based on sensitivity, but also on other main parameters, such as the quality factor (Q.F), figure of merit (FoM), and others, as listed in Table 1. Meanwhile, the sensitivity (S) is known as the ratio of the change in the resonant wavelength $\lambda_{res}$ to the variation in the refractive index of a milky sample, which is measured in nm per RIU such that [82]
(22)$$S=\frac{\Delta \lambda_{res}}{\Delta N}$$

Moreover, the quality factor and figure of merit demonstrate the sharpness of the resonant peak. These two factors are inversely proportional to the full width at half maximum (FWHM) of the resonance peak, such as [80]
(23)$$Q.F=\frac{\lambda_{res}}{FWHM}$$
(24)$$FoM=\frac{S}{FWHM}$$
(25)$$FWHM=\lambda_{L}-\lambda_{R}$$

Furthermore, there are many significant factors such as the signal noise ratio (SnR) which represents the variation in the resonant position, sensor resolution (SnR), detection accuracy (D.A), dynamic range (D.R), and detection limit (D.L) which is inversely proportional to quality factor and uncertainty ($U_{peak}$). As a result, all of these parameters can be enumerated using the following expressions [80]:(26)$$SnR=\frac{\Delta \lambda_{res}}{FWHM}$$
(27)$$D.L=\frac{\lambda_{res}}{20\;S\;Q.F}$$
(28)SR=(D.L) S
(29)$$D.A=\frac{1}{FWHM}$$
(30)$$D.R=\frac{\lambda_{res}}{\sqrt{FWHM}}$$
(31)$$U_{Peak}=\frac{2\Delta \lambda_{res}}{9{\left(SnR\right)}^{0.25}}$$

Below, Table 1 indicates the performance of our designed 1D PC sensor based on the values of different parameters. In this context, the designed sensor provides relatively high values of sensitivity, especially at low fat concentrations. Moreover, our sensing tool provides low values of the order 10^−4^ of the detection limit values. In addition, the small values of the signal to noise ratio can be investigated. Finally, to sum up, Table 2 presents a brief comparison between our considered sensor with some other previous PC sensor structures.

## 4. Fabrication Facilities

In this section, we outline the key aspects of the fabrication procedure for our proposed PC design. The fabrication processes of PC structures have received significant attention due to their contributions to various fields in recent decades. Our design primarily relies on two categories of metamaterials: GMM and HMM. In this context, several techniques are available for fabricating a tri-helix configuration of GMM, including electron beam lithography, spin coating, electrochemical etching, and low-pressure chemical vapor deposition (LPCVD), which involve embedding a metallic layer within a hosting medium [91,92]. Initially, we deposited an ultra-thin layer of silver (Ag) within the dielectric material of titanium dioxide (TiO₂) using various techniques, including low-pressure chemical vapor deposition (LPCVD), RF sputtering, self-assembly, and magnetron sputtering to create the gyroidal metamaterial (GMM) layer. Next, the front face of the GMM can be coated with a polyimide polymer film, which provides exceptional mechanical properties, excellent optical transparency, and high heat resistance. Subsequently, the second type of metamaterial, HMM, can be fabricated using some methods such as electron-beam vacuum evaporation or spin coating techniques. Notably, the first layer of the HMM is composed of porous SiC, which can be fabricated using the sacrificial template method, which is a highly effective technique for creating controllable pore structures [93,94]. The performance of porous SiC layers is influenced not only by the physical and chemical properties of silicon carbide itself but also by pore characteristics, such as porosity. We noted that Wang et al. utilized the liquid infiltration method to control the pores in SiC [95,96]. After that, we precipitated a thin layer of silver metal with a number of periodicity (m) to manufacture the HMM layer. This precipitation process makes the whole composite structure appear as in this configuration [*G*(*ED*)*^m^*]*^S^*. Then, the reflectivity of the structure can then be measured using Fourier transform infrared (FTIR) spectroscopy. Finally, it is worth noting that the experimental validation of PC structures based on many different materials has received considerable interest during the past two decades. For instance, Zhang et al. prepared a 1D PC structure composed of Si and SiO_2_ layers with low ultra-emissivity in NIR based on electron beam coating technology [97]. They investigated an average emissivity of 0.076 which is considered very low compared to those of conventional metal films. In addition, Qi et al. utilized the magnetron sputtering method to prepare the 1D Ge/ZnS PC structure comprising three-unit cells [98]. The fabricated structure exhibited a wide PBG in NIR with 95.1% reflectivity and a low emissivity of 0.054 as well. Moreover, Gryga et al. utilized an optical filter in visible light using the sputtering deposition technique [99].

Moreover, Shekhar et al. have experimentally investigated a 1D HMM based on the inclusion of thin muti-layers of Au, Ag, or Al_2_O_3_ with TiO_2_ using electron beam evaporation [100]. Furthermore, the study of Davidovich demonstrated the optical characteristics of two types of HMM including (Ag/Al_2_O_3_) and (Ag/LiF) using the magnetron sputtering technique at low-pressure plasma [101]. Recently, Feng Wu et al. introduced a 1D PC composed of six unit cells as an omnidirectional narrow-band filter of angle-insensitive ultraviolet PBG [102]. In this regard, the proposed PC structure comprises two-unit cells of the HMM structure that is mimicked by dielectric–metallic multilayers as (AlN/Ag).

## 5. Conclusions

In summary, we have presented a simple and efficient sensor based on a one-dimensional photonic crystal structure that incorporates gyroidal and hyperbolic metamaterials. The proposed sensor is designed for the detection of fat concentrations in milk samples. The sensor is configured as ${\left[(Ag/TiO_{2}){\left(P(SiC)/Ag\right)}^{m}\right]}^{S}.$ The transfer matrix method, effective medium theory, and the Drude model represent the mainstay of our theoretical framework towards the introduction of the numerical results. Meanwhile, a comprehensive optimization process was carried out to determine the optimal values of various parameters, including the angle of incidence, filling ratio, thickness, and filling fraction, to enhance the sensor’s performance. In this regard, the numerical results demonstrated that the best performance of the designed sensor can be achieved at the optimum values of *p* = 0.20, ff=0.08, h=0.4, $d_{1}=40\;nm$, $d_{2}=80\;nm$, $d_{3}=120\;nm$, *m* = 4, *S* = 2, and $\theta_{0}=0{}^{\circ}$. As a result, the designed structure achieves a relatively high sensitivity of 3864 nm/RIU and a detection limit of about 10^−4^ at the optimal values of these parameters. We believe that our biosensor could be highly beneficial for quality control in food processing industries compared to its counterparts using photonic structures. Finally, we aim in future work to consider some new parameters to improve the performance of the designed sensor, such as fabrication tolerance, temperature and surface roughness as well.

## Figures and Tables

**Figure 1 nanomaterials-14-01734-f001:**
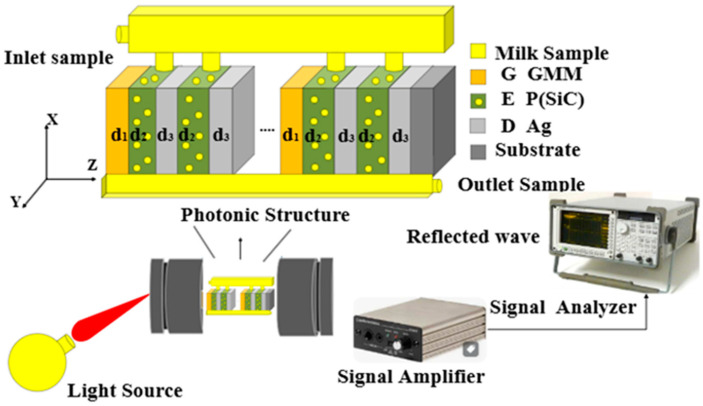
The schematic diagram of the 1D PCs comprising HMMs and GMMs to act as a sensor for the detection of fat concentrations in milk. Here, each layer has a specific thickness and refractive index. Then, the refractive index of air and the substrate is denoted as $n_{0},n_{S}$, respectively.

**Figure 2 nanomaterials-14-01734-f002:**
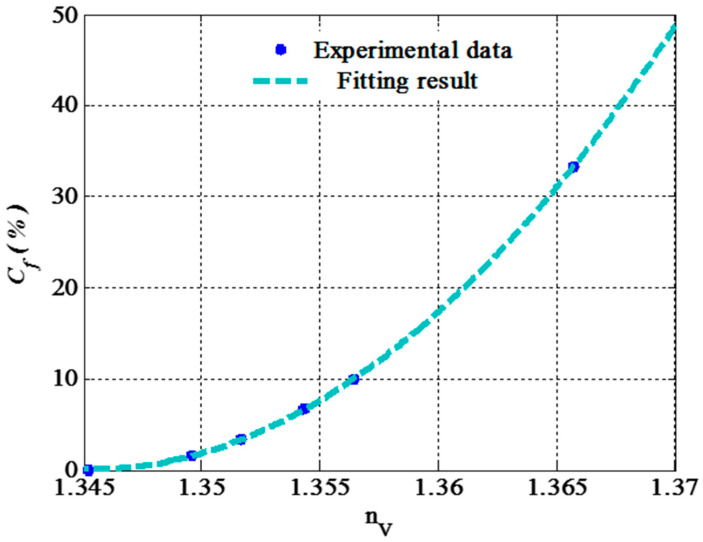
The impact of the fat concentration in commercial milk on its refractive index.

**Figure 3 nanomaterials-14-01734-f003:**
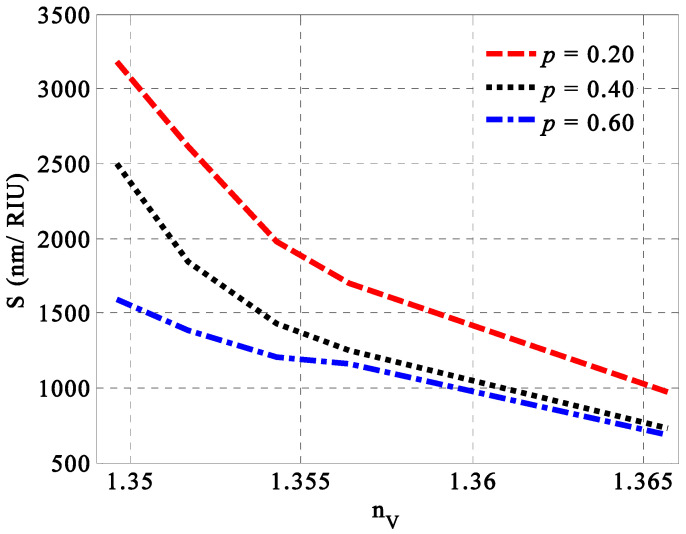
The sensitivity of the designed 1D PC sensor versus the refractive index of milk samples due to the changes in their fat concentrations at three different values of SiC layers’ porosity, i.e., *p* = 0.2, 0.4, and 0.6.

**Figure 4 nanomaterials-14-01734-f004:**
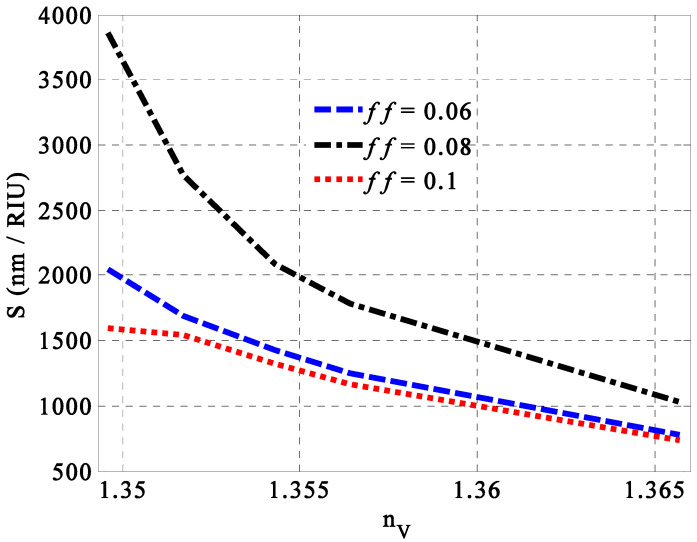
The impact of the Ag filling fraction through GMM layers on the sensitivity of the designed sensor at different concentrations of fats in milk samples.

**Figure 5 nanomaterials-14-01734-f005:**
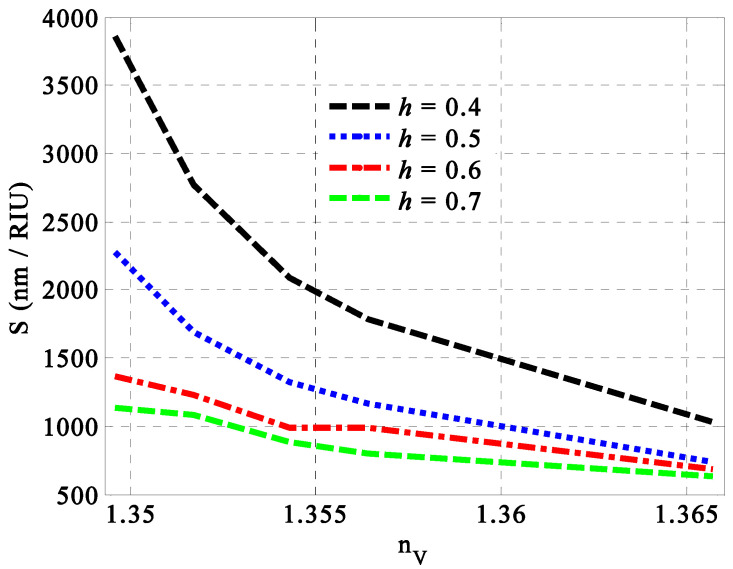
The response of the sensitivity of the designed sensor versus the filling ratio of the HMM layers at different concentrations of fats in the milk samples.

**Figure 6 nanomaterials-14-01734-f006:**
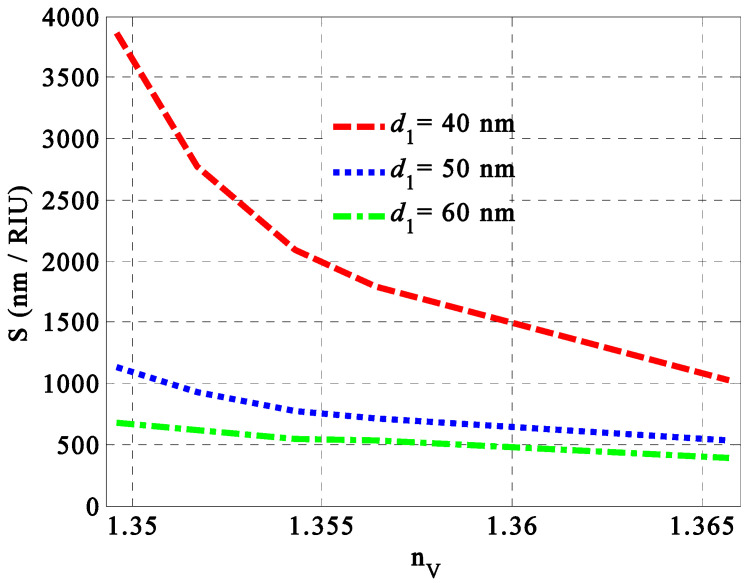
The diagram demonstrates the perfect Gyroidal thickness $d_{1}$ used to obtain higher sensitivity in our proposed sensor.

**Figure 7 nanomaterials-14-01734-f007:**
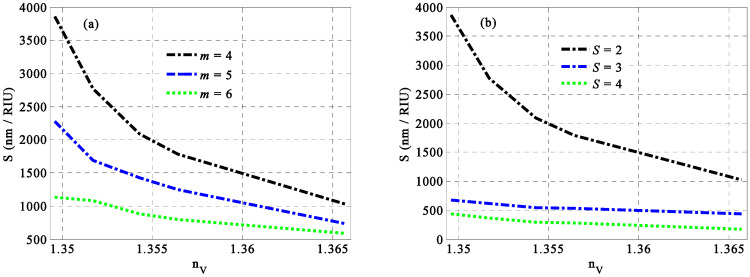
The sensitivity of our ${\left[(Ag/TiO_{2}){\left(P(SiC)/Ag\right)}^{m}\right]}^{S}$ sensor versus different values of the refractive index of the milk sample for a certain number of periods of (**a**) HMM layers and (**b**) the whole structure.

**Figure 8 nanomaterials-14-01734-f008:**
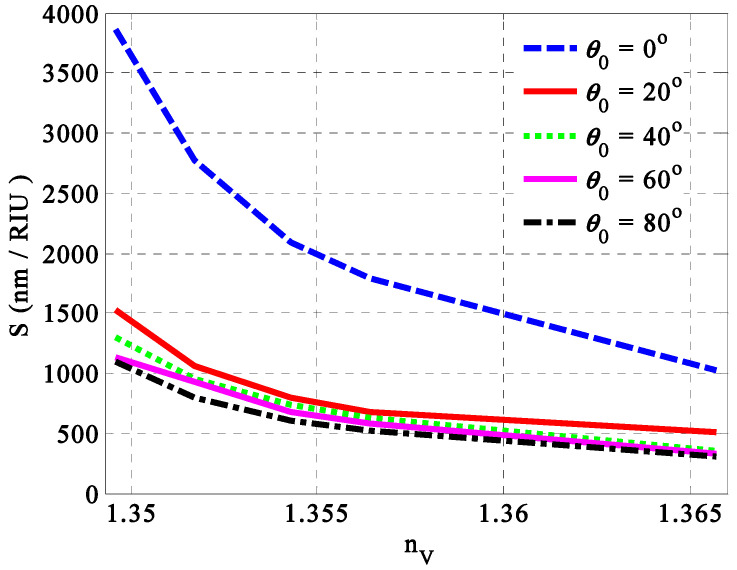
The sensitivity versus the concentration of fats in milk samples at specific angles of incidence.

**Figure 9 nanomaterials-14-01734-f009:**
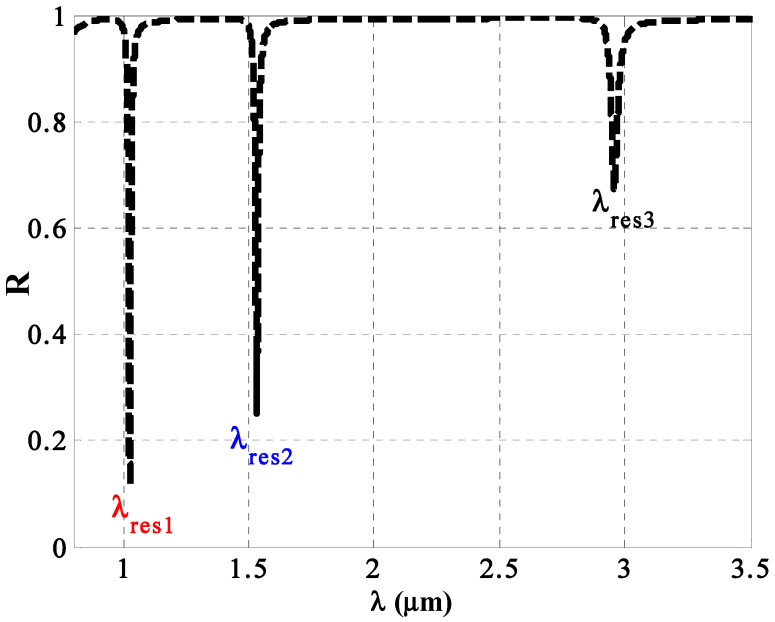
The reflectance spectra of the ${\left[(Ag/TiO_{2}){\left(P(SiC)/Ag\right)}^{4}\right]}^{2}$ sensor regarding the incident wavelengths for a normal incidence case.

**Figure 10 nanomaterials-14-01734-f010:**
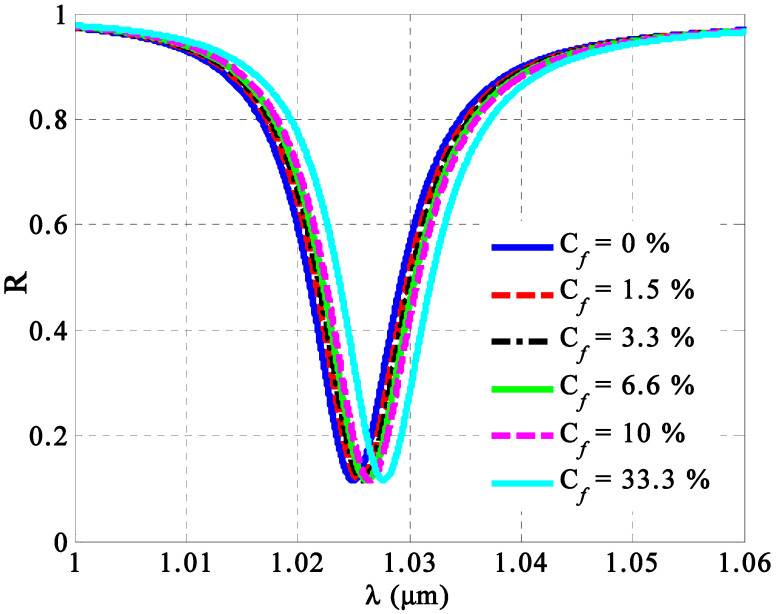
The response of the TP resonance peaks against the changes in the fat concentrations of milk samples.

**Table 1 nanomaterials-14-01734-t001:** The performance of the 1D PC sensor at different fat concentrations in the considered milk samples.

Parameter						
$n_{S}$ (RIU)	1.3452	1.3496	1.3517	1.3543	1.3564	1.3657
$C_{f}$ (%)	0	1.5	3.3	6.6	10	33.3
Position $\lambda_{res}$ (μm)	1.016	1.033	1.034	1.035	1.036	1.037
FWHM (μm)	0.008	0.008	0.008	0.007	0.007	0.007
S (nm/RIU)	-	3863.64	2769.23	2087.91	1785.71	1024.39
Q.F	127	129.13	129.25	147.86	148	148.14
FoM (RIU^−1^)	-	482.95	346.15	298.27	255.1	146.34
SnR	-	2.13	2.25	2.7142	2.86	3
D.A (nm)^−1^	0.125	0.125	0.125	0.14	0.143	0.14
D.L (RIU)	-	0.0001035	0.000144	0.0001676	0.000196	0.0003416
S.R (nm)	-	0.4	0.4	0.35	0.35	0.35
DR	11.36	11.55	11.56	12.37	12.38	12.39
$U_{peak}\;(nm)$	-	3.13	3.27	3.29	3.42	3.55

**Table 2 nanomaterials-14-01734-t002:** A brief comparison between our considered sensor and some of the previously designed structures.

Ref.	S (nm/RIU)	D.L (RIU)	The Designed Structure	Year
[83]	970	-	Metallo-dielectric heterostructure configuration based on TP resonance	2015
[84]	1179	~2.2 × 10^−5^	Photonic crystal containing graphene	2016
[85]	1118	10^−3^	Photonic crystal biosensor	2016
[86]	450	~1.6 × 10^−4^	Photonic crystal cavity and fiber loop ring-down technique	2016
[87]	17	-	Photonic crystal covered with a perforated gold film	2017
[88]	777	-	Cavity photonic crystal	2020
[29]	655.34	-	1D gyroidal PCs with a defect layer	2022
[89]	98.093	-	1D heterostructure PCs	2021
[2]	350	-	Detection of fat concentration in milk using SPR biosensor based on Si and Ti_3_C_2_T_x_	2022
[90]	400	-	Performance analysis of photonic crystal-based biosensor for the detection of biomolecules in urine and blood	2023
Our design	3863.63	1 × 10^−4^	Our designed structure, ${\left[(Ag/TiO_{2}){\left(P(SiC)/Ag\right)}^{4}\right]}^{2}$	2024

## Data Availability

The datasets used and/or analyzed during the current study are available from the corresponding author on reasonable request.

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
