# Peer review of "One-Dimensional Photonic Crystals Comprising Two Different Types of Metamaterials for the Simple Detection of Fat Concentrations in Milk Samples"

_nanomaterials, 2024, doi:10.3390/nano14211734_

Round 1

Reviewer 1 Report

Comments and Suggestions for Authors

An interesting work, however I figure some revision should be made before acceptance for this journal. The first item is, the size range of fat globules in homogeneous milk is 0.2-2 µ m, how can the design be suitable for flat concentrations detection? What type of milk can it work for? The second item is, in figure 1, the schematic diagram of the 1D PCs can not represent clearly the design detail, the 3D diagram with size details should be shown there. 

Comments on the Quality of English Language

The introdution part should be improved, so as to focus on the detection of fat concentrations in milk samples, not the milk itself.

Author Response

Thanks a lot for your great efforts in reviewing our paper. We respond to all of your comments and recommendations through the attached Authors' response Letter.

Reviewer 2 Report

Comments and Suggestions for Authors

This paper is a practical application of photonic crystals based on multilayers of a complex materials. Authors offer a detailed studi of the influence of different parameters in the performace of these sensors.

As milk is a chemically very complex matrix the sensitivity sould be high to detect and quantify other chemical species in it. The detection of fat in milk in the proposed structures profits from the high content in fat and on the important difference in dielectric constant between this and water.

Author Response

(The authors gave the same response as above.)

Reviewer 3 Report

Comments and Suggestions for Authors

The authors introduce a multilayer design, referred by them as a 1D photonic crystal, for detecting fat concentration in milk samples. By reading the Abstract, Introduction and Conclusion sections of the paper, it is not obvious that this design has yet to be constructed, and only numerical simulations have been conducted so far. According to the reviewer, neither the proposed design nor the simulations adequately demonstrate the viability of the device as a sensor. The scientific content of the article contains significant deficiencies, and further experiments and simulations are necessary before it can be considered for any scientific publication. Consider the following:

1. Figure 1 presents the schematic diagram of the proposed sensor. At the top of the proposed design, a reservoir is present where milk samples can be introduced through an inlet, while an outlet is also provided. Below this reservoir, a metal-dielectric multilayer structure is placed, with some dielectric layers made of porous SiC. The detection mechanism depends on the infiltration of the pores by the milk sample. However, it is questionable whether the milk sample can infiltrate the pores in its entire cross-section of the layers. The further question is whether a new milk sample can effectively replace the previous one in the pores. When a subsequent sample is introduced, it may predominantly travel from the inlet to the outlet through the reservoir, with minimal or partial mixing with the prior sample that already occupies the SiC layer pores. As a result, this could lead to unreliable sensor readings.

2. The theoretical description is inadequate. A photonic crystal consists of several unit cells. The proposed structure consists of two unit cells, which usually cannot be considered a photonic crystal. This is clearly visible in the reflectance spectra of Figure 8, which is a spectrum typical of resonant structures and not photonic crystals. There is no information presented on the origin and nature of these resonant modes, of which the first is utilized for sensing.

3. The goals and benefits of assembling two types of 'metamaterials', namely hyperbolic and gyroidal metamaterials, are not discussed. Again, in the same comment as in the previous section, a metal-dielectric-metal or a dielectric-metal-dielectric multilayer cannot be regarded as a metamaterial. Therefore, the applicability of homogenization formulas 9-10, therefore the full design procedure is questionable. 

4. The electric permittivity of milk is essential to assess the data presented, but is not included in the paper. Provide the dependence of the electric permittivity of milk as a function of fat concentration.

5. It is claimed that optimization of several parameters has been performed; see Section 3.1 Optimization of parameters, with subtitles as 3.1.1. Optimization of porosity, 3.1.2. Optimization of filling fraction, etc. However, no optimizations where performed only three parameter sweeps for each case, on the basis of which it is not possible to choose an optimal geometry. For example, in the subtitle of Figure 5. reads 'The diagram demonstrates the perfect Gyroidal thickness used to get higher sensitivity in our proposed sensor.' and in the main text it is concluded that the upper curve for d1 = 40 nm is the optimum. How the sensitivity would be in case of a 30 nm thickness? Would that be more 'perfect'?

Comments on the Quality of English Language

English should be improved. Omit the wording 'Interestingly, ...' .

Author Response

(The authors gave the same response as above.)

Reviewer 4 Report

Comments and Suggestions for Authors

The authors here proposed a simple and efficient sensor based on 1D PC structure comprising gyroidal and hyperbolic metamaterials, and for simple detection of fat concentrations in milk samples. The work is failed in highlighting the novelty and the structure of the manuscript need to be improved to a great extent.

1. Figure 1 schematic illustration of the this work can not show the strategy clearly, please improve it.

2. The authors should provide real image of some of the experimental results, not just provide the data or plots. Current manuscript do not show how the as-proposed photonic crystals sensors really looks like?

3. How is the analytical performance of the detection strategy, such as specificity, reproducibility, etc., is this method could be used to quantitative analysis?

4. Experimental section is missing.

5. Please highlight novelty of this work.

Author Response

(The authors gave the same response as above.)

Round 2

Reviewer 3 Report

Comments and Suggestions for Authors

The authors have made some improvements of the paper, specifically by providing the dielectric constant of milk in function of fat concentration. However the  scientific content of the article still contains significant deficiencies, and further experiments and simulations are necessary before it can be considered for any scientific publication. The paper is misleading, by reading it is not obvious that this design has yet to be constructed, and only numerical simulations have been conducted so far. For example see the modified Fig. 1, which suggest that experiments are performed or the newly added Fabrication facilities paragraph. Fabricate the sensor and provide experimental data that it is superior to similar devices. The theoretical discussion is also inadequate, see the comments of the previous submission, especially those to the benefit of using two type of metamaterials and the origin of resonances, which where not properly addressed. Verify the simulation data with a full wave solver e.g. CST Microwave Studio, HFFS or similar. Conduct microfluidic simulations to ensure that the milk flows properly through the device. Collect the results of the parameter sweeps in a concise form.

Comments on the Quality of English Language

Language and grammar editing is still necessary. 

Author Response

Thanks a lot for your efforts in reviewing our paper. In this regard, we responded to all of your comments, especially some of them are the same as listed in the first round of reviewing. Kindly, find our response through the attached response letter.

Reviewer 4 Report

Comments and Suggestions for Authors

Most of my concerns have been reply well by the authors. Yet for my question 2, I mean to provide some digital photographs of the 1D PC. Could the authors add this in the manuscript?

Author Response

Thanks a lot for your efforts in reviewing our paper. In this regard, we responded to all of your comments. Kindly, find our response through the attached response letter.

Round 3

Reviewer 3 Report

Comments and Suggestions for Authors

The authors have not made considerable improvements of their paper. The comments of the previous two review cycles still holds, they are not properly addressed. The paper in the current form is inadequate for scientific publication.

Comments on the Quality of English Language

The quality of english language should be improved.

Author Response

Actually, we have responded to all of your concerns regarding our paper through the first and second rounds of the revision, as indicated in the attached response letter and the revised version as well. In this regard, we responded to your comments in the first round in red color and in blue color through the second round. Notably, you mentioned in his review report 6 comments, and we responded to all of them as you can note in the response letter. For example, he mentioned in the first round that “The theoretical description is inadequate. A photonic crystal consists of several unit cells. The proposed structure consists of two-unit cells, which usually cannot be considered a photonic crystal.” In this regard, we responded to his comment by plotting the transmittance spectrum of the considered structure at different numbers of unit cells to prove that the photonic band gap can be present at a periodicity number of N = 2 as a result of the superior optical properties of the hyperbolic metamaterial. However, you came back again to repeat the same comment in the second round. Also, you talked about the absence of the experimental verification of the considered structure, and we added a new section regarding the experimental verifications of PC structures and the different methods and techniques used for this purpose. In particular, we have not provided the ability to fabricate this design at this time. However, you repeated this comment again, and we responded as well. Then, you recommended with the language editing, and we did this through the two rounds of revision. However, you are still not satisfied. Finally, for the recommended softwares, we responded that our theoretical and simulation methodology is essentially based on the effective medium theory, characteristic matrix method, Drude model, Bruggeman’s approximation, and Sellmeier formula. Therefore, all of the physical aspects regarding the optical characteristics of the considered materials and the nature of the electromagnetic waves interaction with the designed structure are completely considered. Notably, the transfer matrix method is considered the most efficient method regarding the 1D PC problems and was demonstrated through hundreds of papers on both 1D photonic and 1D phononic designs due to its accuracy, simplicity, and efficiency as well. For the suggested software, unfortunately, we have not received the license for these programs. We hope to introduce some work depending on them parallel to our work with the analytical method (transfer matrix method).
